# Coulomb spin liquid in anion-disordered pyrochlore $Tb_2Hf_2O_7$

Romain Sibille [1,2], Elsa Lhotel[3], Monica Ciomaga Hatnean[4], Gøran J. Nilsen[5,6], Georg Ehlers [7], Antonio Cervellino[8], Eric Ressouche[9], Matthias Frontzek [2,7], Oksana Zaharko[2], Vladimir Pomjakushin[2], Uwe Stuhr[2], Helen C. Walker[6], Devashibhai T. Adroja[6], Hubertus Luetkens[10], Chris Baines[10], Alex Amato [10], Geetha Balakrishnan [4], Tom Fennell[2] & Michel Kenzelmann [1,2]

The charge ordered structure of ions and vacancies characterizing rare-earth pyrochlore oxides serves as a model for the study of geometrically frustrated magnetism. The organization of magnetic ions into networks of corner-sharing tetrahedra gives rise to highly correlated magnetic phases with strong fluctuations, including spin liquids and spin ices. It is an open question how these ground states governed by local rules are affected by disorder. Here we demonstrate in the pyrochlore $Tb_2Hf_2O_7$, that the vicinity of the disordering transition towards a defective fluorite structure translates into a tunable density of anion Frenkel disorder while cations remain ordered. Quenched random crystal fields and disordered exchange interactions can therefore be introduced into otherwise perfect pyrochlore lattices of magnetic ions. We show that disorder can play a crucial role in preventing long-range magnetic order at low temperatures, and instead induces a strongly fluctuating Coulomb spin liquid with defect-induced frozen magnetic degrees of freedom.

[1] Laboratory for Scientific Developments and Novel Materials, Paul Scherrer Institut, 5232 Villigen PSI, Switzerland. [2] Laboratory for Neutron Scattering and Imaging, Paul Scherrer Institut, 5232 Villigen PSI, Switzerland. [3] Institut Néel, CNRS–Université Grenoble Alpes, 38042 Grenoble, France. [4] Physics Department, University of Warwick, Coventry CV4 7AL, UK. [5] Institut Laue-Langevin, CS 20156, 38042 Grenoble, France. [6] ISIS Facility, STFC Rutherford Appleton Laboratory, Chilton, Didcot OX11 0QX, UK. [7] Quantum Condensed Matter Division, Oak Ridge National Laboratory, Oak Ridge, TN 37831, USA. [8] Swiss Light Source, Paul Scherrer Institut, 5232 Villigen PSI, Switzerland. [9] Université Grenoble Alpes, CEA INAC, MEM, 38000 Grenoble, France. [10] Laboratory for Muon Spin Spectroscopy, Paul Scherrer Institut, 5232 Villigen PSI, Switzerland. Correspondence and requests for materials should be addressed to R.S. (email: romain.sibille@psi.ch)

**M**aterials that evade magnetic order although their constituent spins interact strongly are of high interest in condensed matter physics[1,2]. They can host quantum-entangled spin liquid ground states with hidden topological orders and unusual excitations[3–6]. Several mechanisms are known to avoid magnetic order, such as in low-dimensional magnets or in geometrically frustrated magnets. The concerned materials are characterized by crystal structures imposing topological constraints on the lattice of spins. For instance, in frustrated magnets such as the $A_2B_2O_7$ rare-earth pyrochlores[7], the ground state arises from local rules that govern the spin correlations on a single tetrahedron[8–10]. This leads to remarkable spin liquid phases, for instance the dipolar spin ice, where the spin correlations make the system a magnetic Coulomb phase and give rise to emergent magnetic monopole excitations. In that case, strong crystal-field effects constrain large magnetic moments to point along the tetrahedron axes. These moments define classical Ising variables that completely account for the ground state properties in terms of an emergent gauge field. Although most quantum effects are rapidly suppressed with increasing spin quantum number in low-dimensional magnets, in frustrated magnets, phases with strong spin correlations and quantum fluctuations but no long-range order are expected even for large magnetic moments. Several theoretical proposals have been made in the last few years on how to stabilize quantum-entangled phases in pyrochlore magnets[11–17].

It is well established that while strong disorder in dilute systems typically leads to a spin glass, reduced levels of disorder in dense model systems of correlated spins can lead to qualitatively different phenomena. Small amounts of bond disorder in condensed systems can, e.g., disrupt long-range directional coherence by domain formation[18]. In the $S = 1$ antiferromagnetic chain, which features a spin liquid ground state, diamagnetic impurities lead to end-chain cooperative $S = 1/2$ particles reflecting the ground state spin correlations[19–21] – a quantum excitation not present without disorder. In low-dimensional quantum spin systems, bond disorder has been predicted to give rise to random singlet phases[22–25]. In materials where the spin Hamiltonian is frustrated, magnetism is expected to be strongly sensitive to even slight structural changes because the effect of minor perturbations is not preempted by other strong instabilities such as long-range ordering in a conventional magnet. In addition, the crystal lattice can strongly couple to a spin liquid, e.g., through magnetoelastic modes[26] that survive small levels of structural disorder[27] while these are able to completely modify the magnetic ground state[28].

The effect of defects in pyrochlore[29] magnets is only little understood, although it is well established that a number of them feature spin-glass freezing at low temperatures that may be defect-induced[30]. Theoretically, it was predicted that bond disorder on the pyrochlore lattice induces spin glass behavior at very small concentrations[31]. Alternatively, based on experiments, it was argued that missing spins lead to spin freezing controlled by diamagnetic defect-induced states in the spin-liquid background that is not related to the density of defects[32]. Recently, Sen and Moessner proposed that in pyrochlore magnets such defects may lead to the emergence of degrees of freedom with emergent interactions[33]. These can stabilize frozen spin states reflecting the topology of the underlying spin liquid. Most recently, Savary and Balents predicted that for non-Kramers ions, structural defects that preserve the corner-sharing tetrahedral network of magnetic ions can stabilize a disorder-induced quantum-entangled phase at low temperatures[34]. All these studies illustrate a plethora of ground states and excitations in highly frustrated magnets that can emerge from disorder.

Here, we show that anion disorder can be present at a high concentration and potentially tuned in materials where the magnetic ions reside on a perfect pyrochlore lattice. This provides perspectives for systematic studies of the effect of disorder on the rich variety of magnetic ground states stabilized by rare-earth pyrochlore oxides. We present the first study of a pyrochlore magnet with anion Frenkel disorder, $Tb_2Hf_2O_7$, especially using single-crystal samples. We find power-law spin correlations indicating a magnetic Coulomb phase[9] despite local structural defects that affect half of the $Tb^{3+}$ ions with a missing oxygen anion in their coordination environment. This demonstrates a highly correlated state governed by topological constraints that is remarkably robust against the introduction of bond disorder and sites with locally broken symmetry. Further, the power-law spin correlations are even observed in a spin glass phase, demonstrating their coexistence. Finally, our results suggest both a frozen and a fluctuating moment fraction, as in predictions of disorder-induced quantum entangled[34] and topological spin glass[33] phases.

## Results

**Crystal chemistry.** There are two main types of defects in $A_2B_2O_7$ pyrochlore oxides[35]. Antisite defects occur when the A and B cations exchange their positions. Frenkel pair defects occur when oxygen anions are located in an interstitial site accompanied by an oxygen vacancy. The pyrochlore is a charge ordered structure of ions and vacancies and the two types of defects are related to its transformation into a defective fluorite, in which the cations and anions/vacancies are each fully disordered (Fig. 1a, b).

The pyrochlore phase of terbium hafnate, $Tb_2Hf_2O_7$, is located at the border between the stability fields of the pyrochlore and defective fluorite phases[10,29]. This is because the ratio between the ionic radii of the cations, $r_A/r_B$, is just at the limit where the two cations become disordered and form a defective fluorite (hafnates with $r_A > r_{Tb}$, i.e., from $Ce^{3+}$ to $Gd^{3+}$, crystalize in a pyrochlore form). This has two consequences for $Tb_2Hf_2O_7$. First, the material undergoes an order–disorder phase transition towards a defective fluorite phase above 2150 °C[36]. Second, there is a high density of quenched defects in the low-temperature pyrochlore phase. However, although the presence of structural disorder is known, the exact nature of the crystal chemistry of $Tb_2Hf_2O_7$ is a complex question that remains unclear[37–39]. In particular, the antisite cation disorder was never quantified, because usual diffraction methods do not provide enough contrast between Tb and Hf cations, both using X-rays or neutrons.

We combined resonant X-ray and neutron powder diffraction to study the type and density of structural defects in $Tb_2Hf_2O_7$ (Fig. 2). In Fig. 2a–c we present the final joint Rietveld refinement of three powder diffraction patterns that provide a simultaneous sensitivity to cation and anion disorder in $Tb_2Hf_2O_7$. In particular, the diffraction pattern measured near the $L_3$ edge of Tb (Fig. 2a) provides an enhanced contrast between Tb and Hf cations. Our results show that the crystal chemistry of $Tb_2Hf_2O_7$ corresponds to a perfect pyrochlore arrangement of the cations (absence of antisite cation defects, Fig. 2d) with a sizeable density of quenched oxygen Frenkel pair defects (Fig. 2e). In the average structure of our samples of $Tb_2Hf_2O_7$, one of the two oxygen sites that are normally fully occupied in pyrochlore materials ($48f$) is $8 \pm 0.5$ % empty, exactly compensated by the vacancy position ($8a$) occupied at $49 \pm 3$ % (Fig. 1b, c). The accuracy on the antisite cations defect concentration, $0.2 \pm 1.9$ %, is comparable with the limits obtained for $Tb_2Ti_2O_7$ using X-ray absorption fine structure[40].

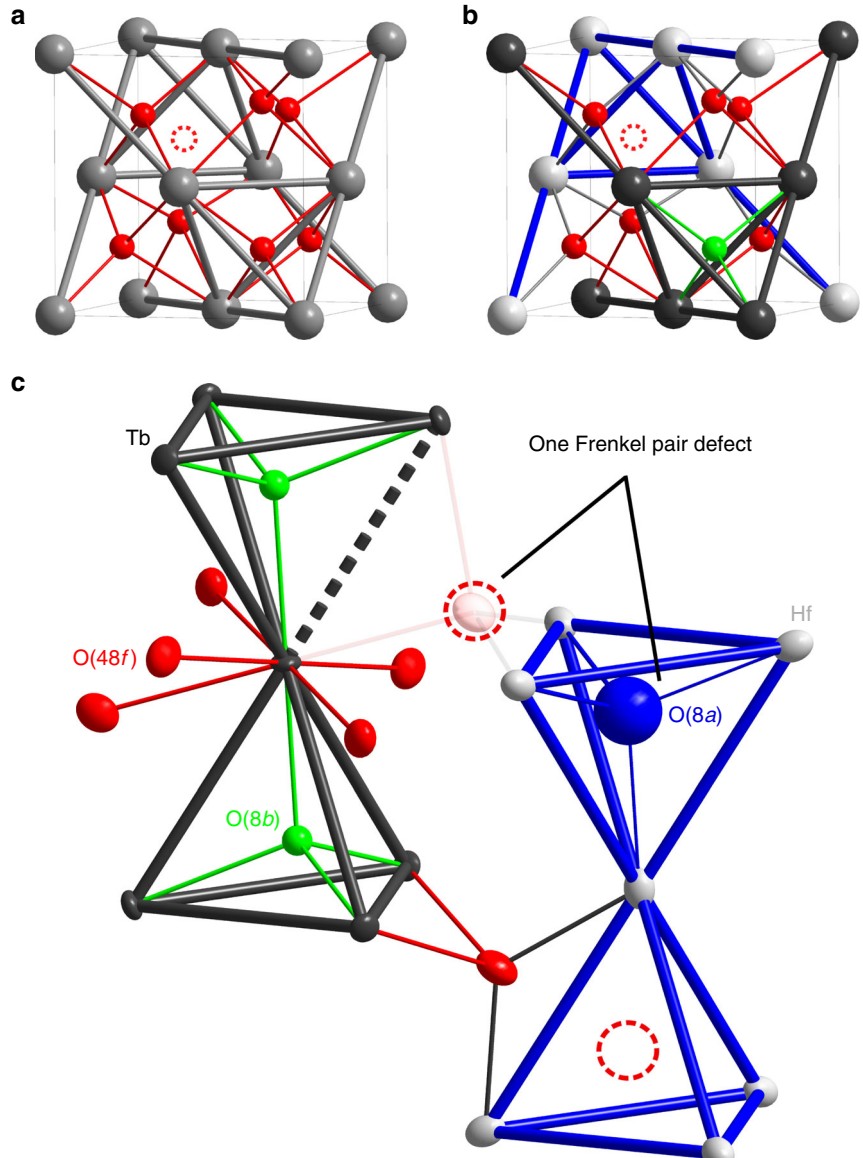

**Fig. 1** Crystal chemistry of defective fluorites, pyrochlores and $Tb_2Hf_2O_7$. In **a**, **b** each unit cell contains one $A_2B_2O_7$ formula unit, but the actual unit cell in **b** is eight times larger than what is displayed. The A (*black*) and B (*light gray*) cations are fully ordered on distinct crystallographic sites in pyrochlore **b** and $Tb_2Hf_2O_7$ **c** structures, while they are disordered on one site (shown in *dark gray*) in the defective fluorite **a**. Ordered cations imply three distinct Wyckoff positions for anions in the pyrochlore structure *Fd*-3*m* **b**: 48 *f* (*red balls*), 8*b* (*green balls*), and 8*a* that is vacant (center of the B-site tetrahedron, 'occupied' by a *red dashed circle* representing a vacancy). Oxygen anions (*red* and *green balls*) and vacancies (*red dashed circles*) can be disordered (**a**), ordered (**b**), or partially disordered (**c**). On average $Tb_2Hf_2O_7$ **c** has ~8% of the 48 *f* positions that are vacant, exactly compensated for stoichiometry by ~50 % of 8*a* positions that are occupied (*blue balls*). This average structure corresponds to a random distribution of oxygen Frenkel pair defects. Such a local defect is shown at the top of **c**, where an empty 48 *f* position is compensated by a non-vacant neighbor 8*a* position, while the situation normally occurring in a perfect pyrochlore is shown at the bottom. $Tb_2Hf_2O_7$ is characterized by a distribution of the two configurations shown respectively on the top and bottom of **c**. A Frenkel defect has two consequences for the magnetic pyrochlore sublattice of $Tb^{3+}$ ions shown in the left part of **c** (*dark gray*). First, the first-neighbor symmetry around ~50 % of the $Tb^{3+}$ cations is broken, because they have seven instead of eight anion ligands. Second, ~8% of the bonds defining the magnetic pyrochlore lattice have one instead of two Tb–O–Tb superexchange pathways, their strength being therefore modified

Frenkel pair defects have two main effects on the magnetism (Fig. 1c): first, they break the local crystal-field symmetry and thus affect the local magnetic moments, and second, they break one of the two magnetic superexchange pathways bridging the first-neighbor magnetic A cations, thus leading to bond disorder. Assuming a random distribution of the oxygen Frenkel pairs, the observed amount of disorder translates into a defective crystal-field environment for ~50% of the $Tb^{3+}$ ions, while ~8% of the magnetic bonds of the pyrochlore lattice are disordered (one bond in every second terbium tetrahedron).

**Evidence for a Coulomb phase**. We now turn to the description of the magnetic properties of the anion-disordered pyrochlore $Tb_2Hf_2O_7$. Figure 3a shows the result of an experiment using time-of-flight neutron spectroscopy and single-crystal samples presenting the same type and density of defects as our powder

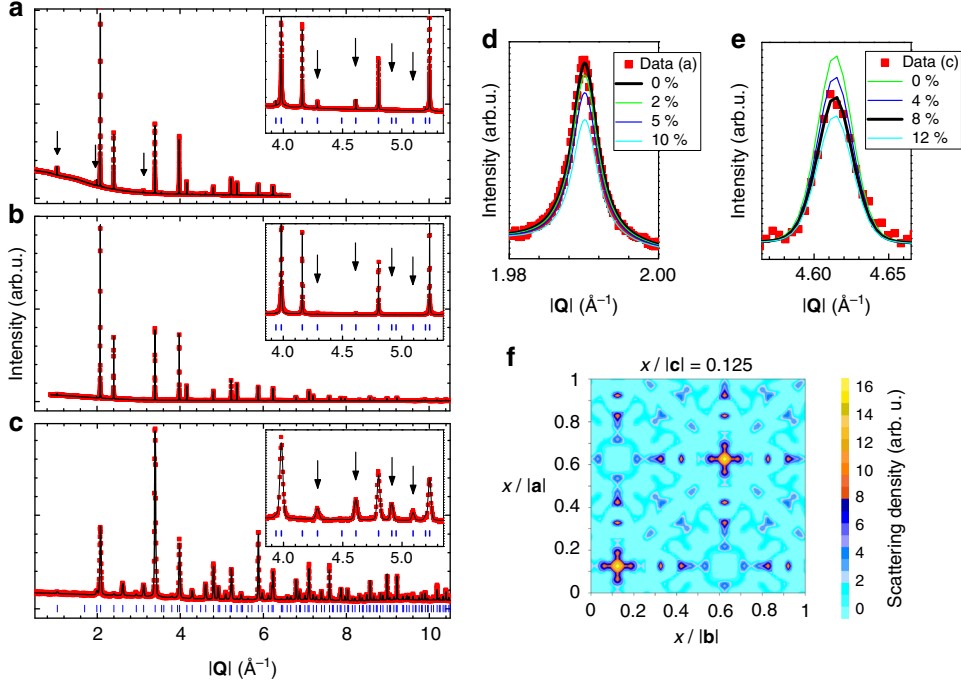

**Fig. 2** Crystallographic characterization of $Tb_2Hf_2O_7$ from powder samples. **a–c** Final Rietveld plots of the joint refinement of high-resolution synchrotron X-ray **a**, **b** and neutron **c** powder diffraction data. The X-ray patterns were recorded using incident wavelengths of **a** 1.65304 Å (i.e., about 10 eV lower than the $L_3$ X-ray absorption edge of Tb, which provides a significant contribution of the anomalous terms to the X-ray scattering factor of Tb) and **b** 0.49599 Å (an energy that minimizes the absorption). The neutron pattern **c** was measured with a wavelength 1.155 Å on a large angular range detector, which yields an excellent sensitivity to the thermal displacement factors in general and to the occupancy factors of the different oxygen sites in particular. *Arrows* indicate some of the satellite reflections originating from the pyrochlore superstructure relative to a defective fluorite phase due to both the disorder-free long-range order of Tb(16$d$) and Hf(16$c$) cations, and to the partially disordered long-range order of oxygen atoms in 8$b$ sites and oxygen vacancies in 8$a$ sites. **d**, **e** show the pyrochlore superstructure reflections (311) and (553) from the data of **a**, **c** respectively, compared to calculations for various densities of antisite cation disorder (**d** the percentage indicates the fraction of Tb cations occupying the Hf site and vice versa) and anion Frenkel disorder (**e** the percentage indicates the fraction of vacancies on the 48$f$ position). **f** Defects as they appear in a difference Fourier map plotted in fractional coordinates $x$ of the crystallographic directions of the cubic unit cell. The map is obtained from the high-resolution neutron powder diffraction data in the case of a refinement to a perfect pyrochlore model (i.e., without oxygen Frenkel pair defects). Additional scattering density is clearly observed around the 8$a$ position (0.125, 0.125, 0.125), i.e., at crystallographic positions that are normally vacant in pyrochlore materials

samples. At the low temperature of this experiment ($T = 1.7$ K), the diffuse magnetic scattering typical of a spin liquid is observed, without any magnetic Bragg peaks that would indicate long-range ordered magnetic moments. The main contribution to the diffuse magnetic scattering is within 0.2 meV of the elastic line, showing that the spin liquid correlations are static on a time scale longer than about 5 ns. Neutron powder diffraction patterns recorded as a function of temperature indicate that this wave-vector dependent diffuse magnetic neutron scattering develops below approximately $T = 50$ K (Fig. 3b). This observation corroborates the development of spin–spin correlations suggested by our magnetic susceptibility $\chi$ and specific heat $C_p$ measurements (Supplementary Fig. 1). The amplitude of the diffuse magnetic scattering grows with decreasing temperature at the expense of paramagnetic scattering, and reaches a plateau at around $T = 1$ K. This magnetic signal reflects magnetic correlations that exist down to at least $T = 0.07$ K. It is revealing to compare our results with a simple model of isotropic spins antiferromagnetically correlated on a single-tetrahedron[41] (see calculated pattern on Fig. 3a and *red line* in the inset of Fig. 3b). The model reasonably accounts for the wave-vector dependence of the magnetic scattering; however, sharper features are clearly visible in the data, which indicates that the magnetic correlations actually extend over many tetrahedra.

A neutron polarization analysis experiment at $T = 0.07$ K (Fig. 3c, d) allows us to distinguish two components of the spin–spin correlation function that are superimposed in Fig. 3a (see Supplementary Fig. 2 for a scheme of the experimental coordinates). The spin-flip and non-spin-flip magnetic scattering maps correspond to correlations among spin components that are parallel and perpendicular to the scattering plane, respectively. Both correlation functions feature sharp and anisotropic scattering in the ($h,h,l$) plane. Fig. 3e shows the line shapes of the two polarization channels along the wave-vectors ($h,h,2$), demonstrating that correlations are much more extended than in the single-tetrahedron model (*red line* on Fig. 3e). The non-spin-flip scattering shows bow-tie features expected for correlations dominated by "2-up-2-down" ice-rule configurations[42], while the spin-flip scattering has sharp and anisotropic features around (0,0,2) and (1,1,0). This highly structured scattering is highly reminiscent of that observed in $Tb_2Ti_2O_7$, where it is associated with a magnetic Coulomb phase[42, 43]. Our neutron pattern has also similar features with those calculated for a recently reported type of Coulomb spin liquid, where the emergent gauge structure is different from that of spin ices[44]. Figure 3f shows the temperature dependence of the scattering at (0.59,0.59,$l$) for three temperatures, showing that while the sharp features build up only below $T = 5$ K, there is very little change of the diffuse scattering between $T = 1.15$ K and $T = 0.07$ K. Therefore, our

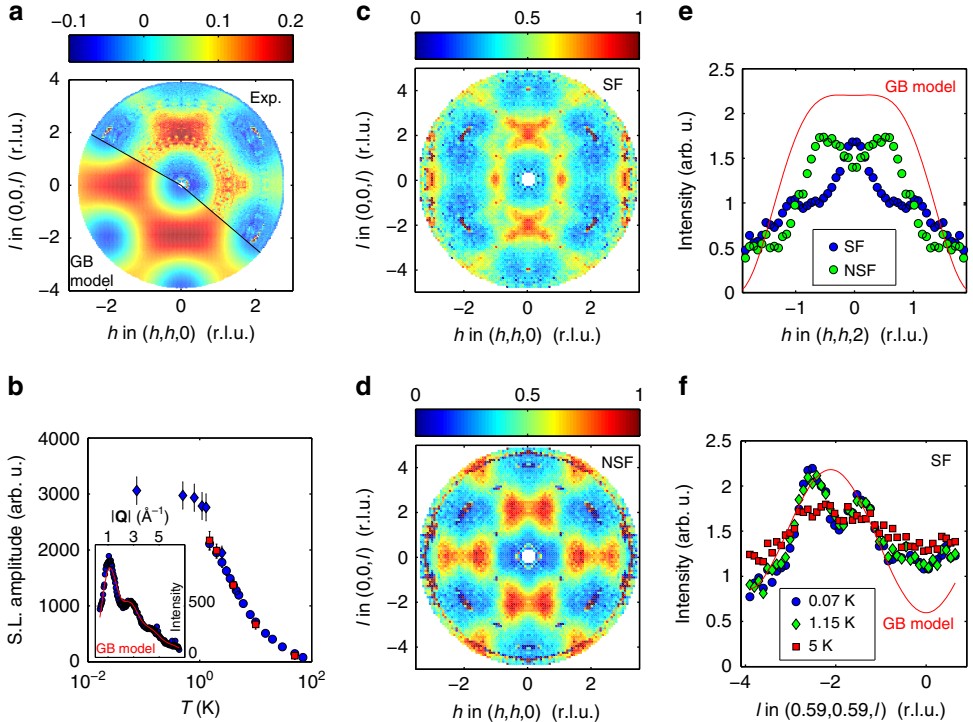

**Fig. 3** Spin correlations in $Tb_2Hf_2O_7$ measured by neutron scattering techniques. **a** Map of reciprocal space in the $(h,h,l)$ plane, in reciprocal lattice units (r.l. u.), of the diffuse magnetic scattering for unpolarized neutrons. Top right (Exp.): single-crystal scattering intensity at $T = 1.7$ K observed with time-of-flight spectroscopy, and integrated over energy transfers [−0.2; 0.2] meV after the subtraction of background scattering determined at $T = 50$ K. Bottom left (GB model): structure factor of an isotropic model of antiferromagnetically correlated spins over a single tetrahedron (the Gardner-Berlinsky model)[40]. **b** Amplitude of the integrated liquid-like magnetic scattering as a function temperature. The inset shows the magnetic scattering at $T = 1.5$ K, fitted with a combination of paramagnetic scattering and the powder average of the structure factor of the Gardner-Berlinsky model[40]. *Red* and *blue points* were measured on two separate instruments, and scaled at $T = 4$ K. The error bars represent the standard deviation of the fit parameter. **c**, **d** In plane/spin flip **c** and out of plane/non-spin flip **d** scattering maps, measured at $T = 0.07$ K using neutron polarization analysis, symmetrized and unfolded. **e** Cuts through the data of **c**, **d** showing the spin flip and non-spin flip scattering of polarized neutrons measured along the wave-vectors $(h,h,2)$. The red curve shows the result of the Gardner-Berlinsky model. **f** Spin-flip scattering of polarized neutrons at three different temperatures along the wave-vectors $(0.59,0.59,l)$, showing the build-up of magnetic correlations below $T = 5$ K, but no change between $T = 0.07$ and $T = 1.15$ K

measurements strongly suggest that around $T = 2$ K $Tb_2Hf_2O_7$ enters a Coulomb phase[9], characterized by power-law correlations, that persists to the lowest temperatures surveyed in our experiment. This Coulomb phase is present despite a high density of defects that affect the crystal-field environment of about half the rare-earth ions and is thus remarkably robust. It is remarkable that a highly disturbed crystal structure gives rise to a low-temperature magnetic phase that resembles the one found in $Tb_2Ti_2O_7$, where the spin-lattice coupling may play an important role[26].

**Evidence for spin glass behavior.** In the magnetic Coulomb phase, an irreversibility in the zero field-cooled (ZFC) – field-cooled (FC) susceptibility curves, $\chi_{dc}$ versus $T$ (Fig. 4a), is observed at $T_{SG} \sim 0.75$ K. This is also visible as an anomaly in our plot of the effective magnetic moment as a function of temperature (Supplementary Fig. 1). Evidence for a macroscopic freezing below $T_{SG}$ is further provided by the real and imaginary parts of the ac-susceptibility, $\chi'_{ac}$ and $\chi''_{ac}$, respectively, measured as a function of temperature for frequencies between $f = 0.57$ Hz and $f = 211$ Hz (Fig. 4a, b). The frequency dependence of the peak temperature $T_f$ in $\chi'_{ac}$ cannot be described by an Arrhenius law and differs from that of a classical spin ice[45]. Instead, the parameter characterizing the shift of $T_f$ with frequency, the so-called Mydosh parameter $\Phi = (\Delta T_f/T_f)/\Delta(\log(f))$, equals

~0.05, which is typical for insulating spin glasses[46], and the frequency dependence of $T_f$ is successfully accounted for by the dynamical scaling law of spin glasses (Fig. 4a, inset), yielding a shortest relaxation time $\tau_0$ of the order of $10^{-8}$ s and a critical exponent $z\nu \sim 7$. In the absence of long-range magnetic order, this provides good evidence that $Tb_2Hf_2O_7$ undergoes a transition to a spin glass at $T_{SG} \sim 0.75$ K. The observation of a canonical spin glass transition in the ac-susceptibility is in remarkable contrast to many frustrated magnets showing frequency dependence of the ac-susceptibility at low temperature, such as in $Pr_2Zr_2O_7$[47, 48] and $Tb_2Ti_2O_7$[49], apart from $Y_2Mo_2O_7$ which is known to enter a spin-glass state below $T_{SG} \sim 22.5$ K[50, 51].

**Temperature dependence of magnetic fluctuations.** The temperature dependence of the spin dynamics is further investigated using zero-field muon spin relaxation histograms measured at various temperatures and fitted by a stretched exponential function $A(t) = A_0 \exp[-(t/T_1)^\beta] + A_{bg}$ (Supplementary Fig. 3). Upon decreasing temperature, the relaxation rate $\lambda = 1/T_1$ increases and the exponent $\beta$ decreases (Fig. 4c). This is the result of the slowing down of spin fluctuations leading to an increase of the width of the field distribution at the muon sites, and of the broad fluctuation spectrum expected in a structurally disordered material. Figure 4c shows that the relaxation rate does not further increase below $T = 10$ K, demonstrating the absence

of further spin slowing down as observed in the classical spin-ice materials[52]. This provides evidence that the spin correlations seen in the diffuse neutron scattering at low-temperature fluctuate on a somewhat faster time scale than the muon Larmor precession frequency (~MHz). Moreover, external longitudinal fields do not modify the relaxation function, showing that the magnetism in $Tb_2Hf_2O_7$ remains dynamic for muons even at the lowest temperature $T = 0.02$ K. In this correlated low-temperature regime, $\beta$ shows a slight increase towards $T = 1$ K, before decreasing again when entering the spin glass phase. While the decrease below $T_{SG}$ is expected because the frozen spins participate in the spin glass, the increased value between $T_{SG}$ and $T = 10$ K shows a slight narrowing of the fluctuation spectrum associated with a highly correlated, fast fluctuating phase. The emergent degrees of freedom freezing below $T_{SG}$ provoke a slight decrease of $\lambda$ but do not lead to an increasing non-relaxing tail in the muon relaxation. This indicates that, below $T_{SG}$, the macroscopically frozen state observed on long time scales coexists with fluctuations at smaller time scales and does not affect the Coulomb phase. This idea is confirmed by neutron scattering measurements in Fig. 3f, which shows that there is no measurable difference in the structure factor above and below $T_{SG}$, while Fig. 3c, d show no sign of diffuse scattering that would be indicative of short-range frozen magnetic correlations. Our results evidence a macroscopically frozen state on slow time scales which coexists with fast fluctuations. It is likely and consistent with our data that, on both the slow and fast time scales, the local rule already imposed at higher temperatures is respected.

## Discussion

Although $Tb_2Hf_2O_7$ has a high density of structural disorder, it has the hallmark of a correlated state governed by a local rule on the pyrochlore lattice – power-law correlations. This shows that highly correlated phases on the pyrochlore lattice are very resilient to defects. Two predicted outcomes are evaded. First, although spin glass formation was predicted for minute densities of defects[31], if this was the case the large density of defects would lead to a spin glass at higher temperature. Second, the formation of a highly correlated magnetic state is surprising because $Tb^{3+}$ is a non-Kramers ion, so in the presence of defects its magnetic doublet states are not protected by time reversal symmetry, and the single-ion ground states are expected to be non-magnetic singlets. Our results suggest that the tendency towards forming a correlated magnetic ground state is stronger and appears to protect the local magnetism, even in a highly asymmetric crystal-field environment.

Our results are consistent with the idea that structural defects around non-Kramers ions stabilize a disorder-induced quantum-entangled phase at low temperatures[34]. In this theory,

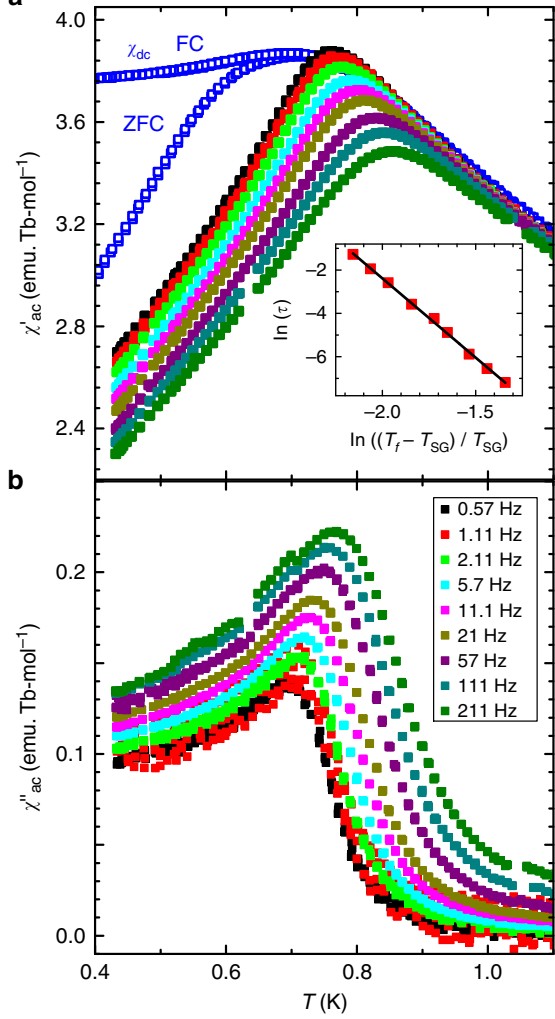

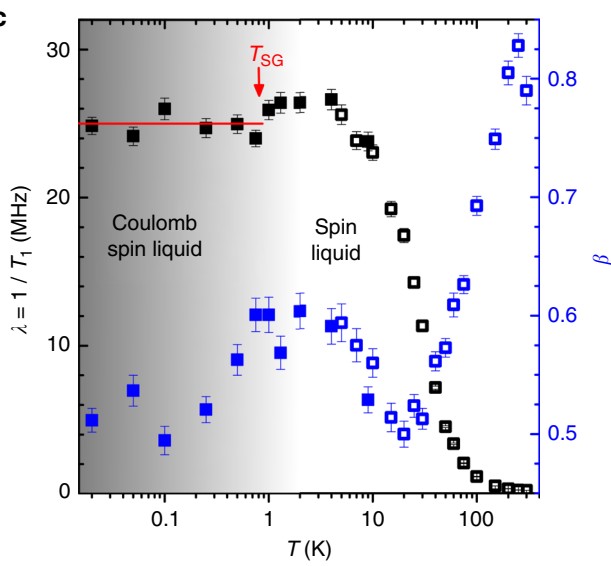

**Fig. 4** Spin dynamics in $Tb_2Hf_2O_7$ probed by ac-susceptibility and muon spin relaxation. **a**, **b** Ac-susceptibility ($\chi_{ac}$) as a function of temperature. The real ($\chi'_{ac}$) and imaginary parts ($\chi''_{ac}$) are shown for several frequencies $f$ of the applied oscillating magnetic field. In **a**, the *blue open symbols* represent the zero-field cooled (ZFC) and field-cooled (FC) dc susceptibility ($\chi_{dc} = M/H$) evaluated from the magnetization ($M$) measured as a function of temperature in a low field ($H = 0.01$ T). The inset in **a** shows $\ln(\tau)$ versus $\ln(t)$, where $\tau = 2\pi/f$ is the characteristic time and $t = (T_f - T_{SG})/T_{SG}$ is the reduced temperature of the peak temperature $T_f$ in $\chi'_{ac}$ at the frequency $f$, with $T_{SG} = 0.68$ K the underlying spin-glass transition temperature. The *black line* is a fit to the dynamical scaling law of spin glasses $\tau = \tau_0 \times t^{-z\nu}$, where $\tau_0 = 4.1 \pm 0.4 \times 10^{-8}$ s is the shortest relaxation time available to the system, $z$ is the dynamic critical exponent, and $\nu$ is the critical exponent of the correlation length ($z\nu = 7.32 \pm 0.06$). **c** Results of zero-field muon spin relaxation experiments as a function of temperature. Spectra shown on Supplementary Fig. 3 were fitted to a stretched exponential function, $A(t) = A_0 \exp[-(t/T_1)^{\beta}] + A_{bg}$. The parameters extracted from the fits are the relaxation rate $\lambda = 1/T_1$ and the exponent $\beta$. The error bars represent the standard deviation of the fit parameters

defects split the non-Kramers doublet ground state of the rare-earth ions by an energy $\Delta$, with a distribution of splitting energies $d\Delta$. From point-charge calculations and our measurements of the crystal-field excitations (Supplementary Fig. 4) we estimate that the splitting of the Kramers levels $\Delta$ is of the order of 0.5–5 K. This is of similar magnitude as the effective interactions, so that the interactions may stabilize a macroscopic wave-function of a quantum spin liquid as proposed by Savary and Balents[34]. Importantly, our calculations and measurements also indicate that the gap to the first excited crystal-field level in $Tb_2Hf_2O_7$ is one order of magnitude higher than $\Delta$, at least 50 K ~ 4.3 meV, so that the ground state doublet split by $\Delta$ is well isolated and describes the physics at low temperature.

An additional effect may arise because of low-energy states around the defects, that are likely to favor correlations that reflect a local interaction rule. For the Coulomb phase of classical spin ice, Sen and Moessner have recently shown theoretically that non-magnetic defects can lead to emergent degrees of freedom and interactions that can stabilize topological spin glass phases[33]. Such a topological spin glass would eventually freeze into a spin-ice configuration. This theory predicts the onset of Coulomb phase fluctuations at some intermediate temperature and a simultaneous appearance of topological spin liquidity and glassiness at lower temperature. We find a similar temperature dependence in $Tb_2Hf_2O_7$, suggesting that it adopts a topological spin glass phase at low temperatures.

We note that it may be possible to study the effect of the concentration of the anion Frenkel disorder on the properties of various pyrochlore magnets by appropriate doping near the border of the stability field. Indeed, it is worth stressing that other pyrochlore magnets also incorporate the same type of anion disorder[53, 54]. Their common feature with $Tb_2Hf_2O_7$ is the presence of a large and weakly electronegative B cation, e.g., $Zr^{4+}$ or $Hf^{4+}$, which can accommodate the high oxygen coordination number needed to stabilize a Frenkel defect. The concentration of Frenkel defects appears proportional to the proximity of the border between the stability fields of the pyrochlore and defective fluorite phases. Also remarkable is the fact that the density of quenched anion Frenkel defects in $Tb_2Hf_2O_7$ should depend on the thermal treatment of the sample, which appears to be the case when comparing our results with those of other reports on the same material[38]. Our experimental approach to investigate defects on the pyrochlore lattice thus provides a basis for future studies where their density may be finely controlled.

Disordered magnetic interactions on top of a geometrically frustrated lattice and the interplay between spin liquidity and glassiness are important problems which need to be experimentally addressed. In $Tb_2Hf_2O_7$ we have demonstrated that a low-temperature state reminiscent of other topological spin liquid phases develops in spite of a high concentration of perturbed superexchange paths. Additionally, we have shown that a highly disordered structure with broken local symmetries around non-Kramers ions does not necessarily hinder the formation of a highly correlated magnetic ground state.

## Methods

**Sample preparation.** The preparation of terbium hafnate powder was carried out by the solid-state reaction of a mixture $Tb_4O_7$ (Aldrich, 99.999%) and $HfO_2$ (Chempur, 99.95%) in air at 1600 °C, providing a brown material whose color was due to $Tb^{4+}$ impurities. A reduction reaction (5% $H_2$ in Ar) at 1000 °C can be used to recover the stoichiometric, white, $Tb_2Hf_2O_7$ polycrystalline material. Single crystals of $Tb_2Hf_2O_7$ were grown by the floating-zone technique using a four-mirror xenon arc lamp optical image furnace (CSI FZ-T-12000-X-VI-VP, Crystal Systems, Inc., Japan). The growths were carried out in air at ambient pressure and at growth speeds of 18 mm h$^{-1}$. The two rods of $Tb_2Hf_2O_7$ polycrystalline material (feed and seed) were counter-rotated at a rate of 20–30

rpm. The crystals were annealed, first in air at 1600 °C for a few days, followed by an annealing in a reducing atmosphere (5% $H_2$ in Ar) for 10 hours at 1000 °C. The resulting transparent crystals were aligned using a Laue X-ray imaging system with a Photonic-Science Laue camera, as well as using the neutron instruments ORION at SINQ, PSI and OrientExpress at the ILL.

**Diffraction experiments.** Diffraction experiments were carried out on white powders of $Tb_2Hf_2O_7$ using the HRPT neutron diffractometer at SINQ, PSI, and the Material Science X04SA synchrotron beamline at SLS, PSI[55]. Samples were enclosed in 6-mm diameter vanadium cans and 0.1-mm diameter quartz capillaries for neutron and synchrotron X-ray experiments, respectively. Three-pattern Rietveld refinements were made using the Fullprof software[56]. We used the resonant contrast diffraction method (see e.g., ref. [57]) in order to enhance the contrast between Tb and Hf. One X-ray diffraction pattern was measured at a resonant energy, near the $L_3$ absorption edge of Tb. The difference in the Tb and Hf dispersion corrections $f'$ at this energy reaches about 18 electrons per atom, providing a high accuracy on the refinement of antisite mixing of the two cations.

Diffraction experiments were also carried out on the grown single-crystals of $Tb_2Hf_2O_7$ on the single-crystal diffractometers D23 (CEA-CRG) at ILL and TriCs at SINQ, PSI. On D23 we used a copper monochromator and $\lambda = 1.28$ Å, while on TriCs it was a germanium monochromator and $\lambda = 1.18$ Å. The results concerning the type and degree of anion disorder in the single-crystals are quantitatively similar to what is obtained from the diffraction experiments on powders, in agreement with the fact that the same annealing sequence was used for both powder and single-crystal materials.

**Macroscopic measurements.** Magnetization ($M$) data were measured in the temperature ($T$) range from 1.8 to 370 K in an applied magnetic field ($H$) of 100 Oe using a Quantum Design MPMS-XL super-conducting quantum interference device (SQUID) magnetometer. Additional magnetization, and ac-susceptibility, measurements were made as a function of temperature and field, from 0.07 to 4.2 K and magnetic fields between 0 and $8 \times 10^4$ Oe, using SQUID magnetometers equipped with a miniature dilution refrigerator developed at the Institut Néel-CNRS Grenoble. Magnetization and susceptibility measurements were corrected for demagnetization effects. The specific heat ($C_p$) of a pelletized sample was measured down to 0.3 K using a Quantum Design Physical Properties Measurement System (PPMS). All macroscopic measurements were carried out on powder samples of $Tb_2Hf_2O_7$. Susceptibility measurements were carried out in addition on a single-crystal of $Tb_2Hf_2O_7$ in order to verify the existence of the spin-glass transition.

**Neutron scattering experiments.** The diffuse magnetic neutron scattering was measured on several instruments. On powder samples this was done using the powder neutron diffractometers HRPT (*red data points* on Fig. 4a) and DMC (*blue data points* on Fig. 4b) at SINQ, PSI. For single-crystal measurements, we used both the time-of-flight instrument CNCS, with $E_i = 3.315$ meV, at the Spallation Neutron Source (ORNL, USA) and the D7 diffractometer installed at the ILL, with $\lambda = 3.1$ Å, using a standard polarization analysis technique with the guide field along the vertical axis [1–1 0]. We used vertical ($z$) polarization analysis (measurement of spin-flip and non-spin-flip scattering) at low temperatures (0.07, 1.15, and 5 K). Additional measurements were realized at high temperature (100 K) in order to carry out a full ($xyz$) polarization analysis (spin-flip and non-spin-flip scattering for each direction of the incoming neutron spin). These high temperature data were also used to normalize the low-temperature magnetic scattering data. This allows to separate the magnetic, incoherent, and coherent nuclear contributions to the total scattering. On CNCS the sample was loaded in a standard helium cryostat while on D7 it was mounted in a copper clamp attached to the cold finger of a dilution insert. The single-crystal samples were of ~2 g for CNCS and 9 g for D7. For both experiments the sample was aligned with the [1–1 0] axis vertical.

An inelastic neutron scattering spectrum was measured on a powder sample using the MERLIN time-of-flight spectrometer at ISIS neutron facility in the UK. In addition, a spectrum with a higher energy resolution but smaller range of energy transfers was recorded on the triple-axis spectrometer EIGER at SINQ, PSI.

**Muon spin relaxation experiments.** Muon spin relaxation (μSR) measurements were performed on a powder sample at the LTF and GPS spectrometers of the Swiss Muon Source at PSI, in the range from 0.02 to 300 K. Muons were longitudinally polarized and spectra were recorded in zero field with earth-field compensation or in applied fields parallel to the beam.

**Data availability.** All relevant data are available from the authors. The datasets for the polarized diffuse scattering experiment on D7 are available from the Institute Laue-Langevin data portal (doi:10.5291/ILL-DATA.5-53-248)[58].

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

## Acknowledgements

We thank C. Paulsen for the use of his magnetometers; P. Lachkar for help with the PPMS; M. Bartkowiak and M. Zolliker for help with the dilution fridge experiments at SINQ; S. Turc for help with the dilution fridge experiment at ILL. The work at the University of Warwick was supported by the EPSRC, UK, through Grant EP/M028771/1. Neutron scattering experiments were carried out at the continuous spallation neutron source SINQ at the Paul Scherrer Institut at Villigen PSI in Switzerland. The μSR experiments were carried out at the Swiss Muon Source SμS at the Paul Scherrer Institut at Villigen PSI in Switzerland. X-ray powder diffraction data were collected at the Materials Science × 04SA beamline of the Swiss Light Source (SLS) synchrotron facility at the Paul Scherrer Institut at Villigen PSI in Switzerland. We acknowledge the Institut Laue Langevin, ILL (Grenoble, France, EU) and the ISIS neutron facility (UK) for the allocated beamtime. This research used resources at the Spallation Neutron Source, a DOE Office of Science User Facility operated by the Oak Ridge National Laboratory. We acknowledge funding from the European Community's Seventh Framework Programme (Grants No. 290605, COFUND: PSI-FELLOW; and No. 228464, Research Infrastructures

under the FP7 Capacities Specific Programme, MICROKELVIN), and the Swiss National Science Foundation (Grants No. 200021_140862 and No. 200021_138018).

## Author contributions

Crystal growth and characterization were performed by R.S., M.C.H., and G.B. Diffraction experiments were carried out by R.S. with A.C., E.R., O.Z., V.P., and M.F. as local contacts. Neutron scattering experiments were carried out by R.S., T.F., E.L., and M.K. with G.J.N., G.E., U.S., H.C.W., and D.A. as local contacts. Muon spin relaxation experiments were carried out by R.S. with H.L., A.A. and C.B. as local contacts. The data were analyzed by R.S., T.F., E.L., and M.K. The paper was written by R.S. and M.K., with feedback from all authors.

## Additional information

**Competing interests:** The authors declare no competing financial interests.

