## [Peer Review File · Nature Communications]

Reviewers' comments:

Reviewer #1 (Remarks to the Author):

The purpose of this paper is clear: to demonstrate how structural disorders affect the magnetic behaviour of the frustrated spin system on pyrochlore network. This is one of the central issues of the magnetism of frustrated systems.

However, the presentation of this manuscript is not structured for the readers to reach the authors conclusions about this important issue.

The manuscript consists of results from various kinds of experimental techniques: as the spectroscopies, resonant X-ray diffraction, neutron diffraction and muon spin relaxation, and as the sample characterisation susceptibility and specific heat. Five figures with 18 panels of experimental result is presented in a short communication paper. The explanation is limited by the length of the manuscript and it is very difficult for the readers to follow the relation between the experimental results presented and the proposals which the authors make.

As an example, four types of defects are presented in Figure 1, and powder diffraction patterns are presented in Figure 2 with some explanation, but it is not clear how they are related. Manuscript Line 94-109 is explaining, but it is not clear how the disorder percentages stated may be read from the experimental result (Fig.2) in which the arrow for the ordered structures is the most showing part.

Magnetic spectroscopies by neutron and muon is the most important part of this manuscript, but in the present form, there are too many results shown for the readers to reach to the authors conclusion. The referee would suggest that the author limit the number of panels for the most important ones.

The referee agrees with the importance of this work, but the way the result is presented should be improved for readability.

Reviewer #2 (Remarks to the Author):

In this manuscript the authors report on the experimental study of powder and single crystal samples of the rare-earth pyrochlore $\text{Tb}_2\text{Hf}_2\text{O}_7$, by means of specific heat, magnetization, neutron scattering, RXS and μSR experiments. The authors find a very large disorder due to the proximity of $\text{Tb}_2\text{Hf}_2\text{O}_7$ in phase space to a defective fluorite —but that the magnetic degrees of freedom are nevertheless not quenched— and that strong spin fluctuations coexist with glassy behavior at low temperatures.

The subject is very interesting and topical as it combines the physics of a quantum spin liquid candidate family and the role of disorder in this context, which has been shown to be particularly interesting. Moreover, the study of this new family of rare earth pyrochlores also feels very fresh in this field. The study seems very careful, the manuscript is generally well written, and even though it might *perhaps* be too early to definitively conclude on a particular scenario in this material, this work is more than likely to spur further experimental and theoretical studies of Hf pyrochlores, disordered pyrochlores and more generally quantum spin liquids. In particular the tunability of the disorder in this material makes it particularly promising, and should allow to explore very exciting phenomena.

Therefore I would recommend publication in Nature Communications.

One question: can the authors comment on other Hf rare-earth compounds in the family, i.e. do

they exist in the pyrochlore form?

Reviewer #3 (Remarks to the Author):

The paper reports an extended experimental investigation on the effect of anion disorder in the highly frustrated pyrochlore magnet Tb₂Hf₂O₇.

The authors have combined resonant X-ray and neutron powder diffraction to exactly characterize the cation and anion disorder in Tb₂Hf₂O₇. The crystal chemistry of this substance corresponds to a perfect pyrochlore arrangement of the cations, but to a sizable density of quenched oxygen Frenkel pair defects. These defects may affect the local magnetic moment by breaking the local crystal field symmetry and lead to bond disorder by breaking the super exchange paths between the magnetic cations.

The authors have performed both macroscopic and microscopic magnetic measurements on powder and single crystal sample to characterize the magnetic properties of this frustrated system with considerable magnetic disorder.

These include dc- and ac susceptibility measurements, specific heat measurement, μ SR measurement and numerous neutron scattering experiments at various facilities.

All these measurements indicate an disorder induced spin glass like state below TSG \sim 0.75K preserving the magnetic short range correlations typical to pyrochlore magnets.

Each experimental results are presented adequately and thus can be regarded as important results to warrant publication.

Yet I have one major concern on the interpretation of the experimental results.

I therefore would like the authors to consider following point prior to the publication.

The authors claim to have observed a spin glass like state in the Coulomb phase of Tb₂Hf₂O₇. I think it is crucial to present convincing arguments for the system being in the Coulomb phase using neutron scattering results. Unfortunately the direct comparison and interpretation of the observed short range magnetic correlations do not go beyond the Gardner-Berlinsky model, which describes the correlations within a single tetrahedron only. The other features indicating for correlations going beyond the single-tetrahedron model is hardly discussed in details (see lines 134 to 139).

I therefore do not see convincing argument from the analysis of polarized neutron data pointing to the 'Coulomb phase'.

The statement in line 141 'Therefore, our measurements clearly demonstrate that around T = 2 K Tb₂Hf₂O₇ enters a Coulomb phase, characterized by power-law correlations' seems to be very arbitrary.

Hence the paper in present form does not provide clear experimental findings characterizing a distinct phase as termed 'Coulomb phase' in fig. 4 e .

I thus would like to suggest the authors to strengthen this aspect to warrant the publication in 'Nature Communications'.

In addition to this, the authors might address another objective of common interest. Namely, if the present finding of inherent anion disorder is characteristic to the Tb₂Hf₂O₇ only due to its vicinity of the disordering transition towards a defective fluorite structure or also relevant to other pyrochlore systems being under intense investigations.

Reviewer #1

The purpose of this paper is clear: to demonstrate how structural disorders affect the magnetic behaviour of the frustrated spin system on pyrochlore network. This is one of the central issues of the magnetism of frustrated systems.

However, the presentation of this manuscript is not structured for the readers to reach the authors conclusions about this important issue.

- We thank Reviewer 1 for his/her evaluation of our work and for stressing that our work addresses a central issue in magnetism.

The manuscript consists of results from various kinds of experimental techniques: as the spectroscopies, resonant X-ray diffraction, neutron diffraction and muon spin relaxation, and as the sample characterisation susceptibility and specific heat. Five figures with 18 panels of experimental result is presented in a short communication paper. The explanation is limited by the length of the manuscript and it is very difficult for the readers to follow the relation between the experimental results presented and the proposals which the authors make.

- We thank Reviewer 1 for pointing out the need for better readability. We substantially revised all figures, and we have reduced the number of figures from previously 5 to now 4. The number of panels in the figures has been partially decreased as well. Less important data are now in the Supplementary Information.

As an example, four types of defects are presented in Figure 1, and powder diffraction patterns are presented in Figure 2 with some explanation, but it is not clear how they are related. Manuscript Line 94-109 is explaining, but it is not clear how the disorder percentages stated may be read from the experimental result (Fig.2) in which the arrow for the ordered structures is the most showing part.

- We thank again Reviewer 1 for pointing out the need for a better presentation of the diffraction results and their relation to the densities of defects.

We substantially revised the two first figures so that it is easier for the reader to understand the average structure of the material, and to see directly the relation between the diffraction data and the defect densities given in the main text.

Magnetic spectroscopies by neutron and muon is the most important part of this manuscript, but in the present form, there are too many results shown for the readers to reach to the authors conclusion. The referee would suggest that the author limit the number of panels for the most important ones.

- We moved the least important data in a Supplementary Information, keeping only the most important panels in the main text.

The referee agrees with the importance of this work, but the way the result is presented should be improved for readability.

- We believe that the Reviewer 1 suggestions have helped producing a revised version with a better readability.

Reviewer #2

In this manuscript the authors report on the experimental study of powder and single crystal samples of the rare-earth pyrochlore $Tb_2Hf_2O_7$, by means of specific heat, magnetization, neutron scattering, RXS and μ SR experiments. The authors find a very large disorder due to the proximity of $Tb_2Hf_2O_7$ in phase space to a defective fluorite—but that the magnetic degrees of freedom are nevertheless not quenched—and that strong spin fluctuations coexist with glassy behavior at low temperatures.

The subject is very interesting and topical as it combines the physics of a quantum spin liquid candidate family and the role of disorder in this context, which has been shown to be particularly interesting. Moreover, the study of this new family of rare earth pyrochlores also feels very fresh in this field. The study seems very careful, the manuscript is generally well written, and even though it might *perhaps* be too early to definitively conclude on a particular scenario in this material, this work is more than likely to spur further experimental and theoretical studies of Hf pyrochlores, disordered pyrochlores and more generally quantum spin liquids. In particular the tunability of the disorder in this material makes it particularly promising, and should allow to explore very exciting phenomena.

Therefore I would recommend publication in Nature Communications.

- We are thankful to Reviewer 2 for his/her very positive evaluation of our work.

One question: can the authors comment on other Hf rare-earth compounds in the family, i.e. do they exist in the pyrochlore form?

- The proximity of $Tb_2Hf_2O_7$ to the border between the stability fields of the pyrochlore and defective fluorite structures implicitly means that, for $B = Hf$, the materials with rare-earth cations larger than Tb^{3+} (Ce-Gd) form in the pyrochlore structure while those with rare-earth cations smaller than Tb^{3+} (Dy-Lu) form in the defective fluorite structure.

In order to make this clear for the reader, we added a short note in the main text, on page 5, third paragraph. We thank Reviewer 2 for pointing out the need for this clarification.

Reviewer #3

The paper reports an extended experimental investigation on the effect of anion disorder in the highly frustrated pyrochlore magnet $Tb_2Hf_2O_7$.

The authors have combined resonant X-ray and neutron powder diffraction to exactly characterize the cation and anion disorder in $Tb_2Hf_2O_7$. The crystal chemistry of this substance corresponds to a perfect pyrochlore arrangement of the cations, but to a sizable density of quenched oxygen Frenkel pair defects. These defects may affect the local magnetic moment by breaking the local crystal field symmetry and lead to bond disorder by breaking the super exchange paths between the magnetic cations.

The authors have performed both macroscopic and microscopic magnetic measurements on powder and single crystal sample to characterize the magnetic properties of this frustrated system with considerable magnetic disorder. These include dc- and ac susceptibility measurements, specific heat measurement, μ SR measurement and numerous neutron scattering experiments at various facilities.

All these measurements indicate an disorder induced spin glass like state below $TSG \sim 0.75K$ preserving the magnetic short range correlations typical to pyrochlore magnets.

Each experimental results are presented adequately and thus can be regarded as important results to warrant publication.

➤ We are thankful to Reviewer 3 for his/her very positive evaluation of our work.

Yet I have one major concern on the interpretation of the experimental results. I therefore would like the authors to consider following point prior to the publication.

The authors claim to have observed a spin glass like state in the Coulomb phase of $Tb_2Hf_2O_7$. I think it is crucial to present convincing arguments for the system being in the Coulomb phase using neutron scattering results. Unfortunately, the direct comparison and interpretation of the observed short range magnetic correlations do not go beyond the Gardner-Berlinsky model, which describes the correlations within a single tetrahedron only. The other features indicating for correlations going beyond the single-tetrahedron model is hardly discussed in details (see lines 134 to 139). I therefore do not see convincing argument from the analysis of polarized neutron data pointing to the 'Coulomb phase'.

The statement in line 141 'Therefore, our measurements clearly demonstrate that around $T = 2 K$ $Tb_2 Hf_2 O_7$ enters a Coulomb phase, characterized by power-law correlations' seems to be very arbitrary.

Hence the paper in present form does not provide clear experimental findings characterizing a distinct phase as termed 'Coulomb phase' in fig. 4 e .

I thus would like to suggest the authors to strengthen this aspect to warrant the publication in 'Nature Communications'.

➤ We thank Reviewer 2 for pointing out that the presentation of our neutron scattering measurements and their interpretation might have been confusing

in the first version of the manuscript. We modified the main text and Fig. 3 in order to make our point clearer.

Firstly, the phrasing in the previous version of the manuscript might indeed have given the impression that we used the Gardner-Berlinsky model to interpret our observations, while we simply wanted to show it as a reference in terms of expected scattering for the case of correlations extending on a single tetrahedron only. We modified certain sentences in the respective paragraph, and we also modified Fig. 3 showing the neutron scattering results. In particular, we now compare this simple model additionally to polarized scattering along wave vectors $(h,h,2)$. This makes it clearer that the Gardner-Berlinsky does not reproduce our data at all and that the correlations are extended over a much longer length scale.

Secondly, we also clarified the phrasing regarding the claim for our system being in a Coulomb phase in the main text. In fact, the result of the polarized neutron scattering experiment on $Tb_2Hf_2O_7$ appears highly reminiscent of that observed in $Tb_2Ti_2O_7$, where its association with a magnetic Coulomb phase has been demonstrated and is widely admitted (see ref. 42 as well as other works on this materials by different authors: Guitteny et al. Phys. Rev. Lett. **111** 087201 (2013), reference newly added in the revised manuscript). In addition, the authors of a recent paper calculate very similar scattering for a new type of Coulomb spin liquid (Benton et al. Nature Commun. 7:11572 (2016), reference also newly added to the revised manuscript). Therefore, we think that our claim that $Tb_2Hf_2O_7$ features a magnetic Coulomb phase at low temperatures can be justified by stating the strong similarity between our data and both the data already published on $Tb_2Ti_2O_7$ and the recent theoretical work by Benton et al..

Nonetheless, following the concern of Reviewer 2, and in order to avoid giving the impression of a ‘very arbitrary statement’, we have changed the wording in the sentence “... our measurements clearly demonstrate...”, where it is now written “... our measurements strongly suggest...”.

In addition to this, the authors might address another objective of common interest. Namely, if the present finding of inherent anion disorder is characteristic to the $Tb_2Hf_2O_7$ only due to its vicinity of the disordering transition towards a defective fluorite structure or also relevant to other pyrochlore systems being under intense investigations.

- This request from Reviewer 3 is particularly interesting and is in fact related to a part of the discussion that was briefly touched in the first manuscript, namely that “it may be possible to the effect of the concentration of the anion Frenkel disorder on the properties of various pyrochlore magnets by appropriate doping near the border of the stability field.”

In the revised manuscript, we have slightly extended this paragraph so that the discussion more clearly answers the reviewer’s interest – which certainly many readers will share – on that important point in terms of perspectives:

“Indeed, it is worth stressing that other pyrochlore magnets also incorporate the same type of anion disorder⁵¹⁻⁵². Their common point with $\text{Tb}_2\text{Hf}_2\text{O}_7$ is the presence of a large and weakly electronegative B cation, e.g. Zr^{4+} or Hf^{4+} , which can accommodate the high oxygen coordination number needed to stabilize a Frenkel defect. The concentration of Frenkel defects appears proportional to the proximity of the border between the stability fields of the pyrochlore and defective fluorite phases.”

In conclusion, to answer directly reviewer’s 3 second question: yes, the presence of anion Frenkel defects is also relevant to other pyrochlore magnets that are under intense investigations (at least $\text{Nd}_2\text{Zr}_2\text{O}_7$ and $\text{Nd}_2\text{Hf}_2\text{O}_7$, see newly added references 51 and 52). Our understanding of the presence of the same type of defects in these two Nd-based materials and in $\text{Tb}_2\text{Hf}_2\text{O}_7$ is based on crystal chemistry arguments that are now explained in the manuscript. We think that the presence of anion Frenkel defects requires combination of two factors, namely a large and weakly electronegative B cation and the proximity from the border of the stability field, making clearer our claim that the defects concentration may be tuned in solid solutions.

REVIEWERS' COMMENTS:

Reviewer #3 (Remarks to the Author):

The authors have responded and addressed the major issues raised by the referees. Thus the paper now warrants publication in Nature Communications.